# ABO Blood Groups, RhD Factor and Their Association with Subclinical Atherosclerosis Assessed by Carotid Ultrasonography

**DOI:** 10.3390/jcm13051333

**Published:** 2024-02-27

**Authors:** Malin Mickelsson, Kim Ekblom, Kristina Stefansson, Per Liv, Emma Nyman, Anders Själander, Ulf Näslund, Johan Hultdin

**Affiliations:** 1Department of Medical Biosciences, Clinical Chemistry, Umeå University, 90187 Umeå, Sweden; 2Department of Research and Development, Region Kronoberg, 35234 Växjö, Sweden; 3Department of Public Health and Clinical Medicine, Medicine, Umeå University, 90187 Umeå, Sweden

**Keywords:** ABO blood group system, RhD factor, carotid intima–media thickness, carotid plaques, atherosclerosis, cardiovascular prevention

## Abstract

**Background:** The ABO blood group system has previously been associated with cardiovascular disease (CVD), where non-O blood group individuals have shown an increased risk. Studies assessing early atherosclerotic disease while also including RhD are few. We aimed to determine whether the ABO and RhD blood groups are associated with subclinical atherosclerosis in a healthy population. **Methods:** We included 3532 participants from the VIPVIZA trial with available carotid ultrasonography results to assess subclinical disease. Information about blood groups was obtained from the SCANDAT-3 database, where 85% of VIPVIZA participants were registered. **Results:** RhD− individuals aged 40 years showed increased carotid intima–media thickness (B 1.09 CI 95% 1.03; 1.14) compared to RhD+ individuals. For ABO, there were no differences in ultrasonography results when assessing the whole study population. However, 60-year-old individuals with heredity for CVD and a non-O blood group had decreased odds for carotid plaques (OR 0.54 CI 95% 0.33; 0.88). **Conclusions:** RhD blood group is associated with subclinical atherosclerosis in younger individuals, indicating a role as a mediator in the atherosclerotic process. In addition, a non-O blood group was associated with decreased subclinical atherosclerosis in individuals aged 60 and with heredity (corresponding to the group with the highest atherosclerotic burden).

## 1. Introduction

Cardiovascular disease (CVD) causes a significant number of premature deaths, and in 2019 alone, it is estimated that 18.6 million people died from CVD [1]. While the prevention strategies and concurrent risk prediction algorithms have undergone several improvements, the disease burden remains high. One of the reasons for this is that some young individuals may falsely be classified as low-risk despite already having extensive atherosclerosis [2] and, during their remaining lifespan, might develop a high risk for CVD [3]. Therefore, it is essential to detect new risk markers associated with subclinical atherosclerosis years before the development of CVD to offer new routes to prevention.

The ABO and RhD blood groups are suggested risk markers that are constant and inherited. According to the ABO blood grouping system, there are four possible blood types: A, B, AB, and O. The classification depends on which antigen (A, B, or both) is carried on the erythrocyte’s cell surface, resulting in blood group A, B, or AB, respectively, while those in blood group O express neither of the antigens [4]. RhD status is determined by the presence or absence of the RhD protein on erythrocytes, providing a negative or positive label (RhD− or RhD+).

The ABO blood group system has previously been associated with several circulatory diseases [5,6,7], and it is stated that non-O blood group individuals are associated with increased incidence of arterial thromboembolic events [8,9,10], such as myocardial infarction (MI), but there are more conflicting results as to whether other forms of CVD are associated [10,11,12]. Few studies are available on the RhD blood group and its role in the pathogenesis of circulatory disease, but some have suggested that there are associations between RhD and dyslipidemia, which is a risk factor for CVD [13,14]. Studies assessing the roles of ABO and RhD in subclinical atherosclerosis as potential risk markers are sparse, and the studies available have primarily included individuals already considered high-risk. In this case, it is proposed that non-O blood groups are associated with a higher severity of atherosclerosis, as assessed by angiography [15,16].

There are various methods available for assessing subclinical structural and functional changes in the arterial bed to identify individuals with increased risk of CVD. Pulse wave velocity is commonly used to capture aortic arterial stiffness [17], and the detection of endothelial dysfunction is possible by flow-mediated dilation [18]. Another commonly used method for the detection and measurement of subclinical atherosclerotic disease is ultrasonography of the carotid arteries. Ultrasonography detects the presence and estimates the extent of atherosclerosis by measuring the carotid intima–media thickness (CIMT) and the presence of plaques. The CIMT and carotid plaques have been shown to predict cardiovascular events and aid in risk reclassification [2,19,20,21].

The study aimed to determine whether the non-O and RhD− blood groups are associated with subclinical atherosclerosis. Another aim was to investigate whether an association could be seen among participants reporting heredity for CVD. We investigated this by examining healthy individuals with carotid ultrasonography and retrieving their blood group from the SCANDAT database.

## 2. Materials and Methods

### 2.1. Study Population

All participants were part of the VIPVIZA trial (ClinicalTrials.gov NCT01849575). VIPVIZA (VIsualiZation of Asymptomatic atherosclerotic disease for optimum cardiovascular prevention) is a pragmatic randomized controlled trial nested within the Västerbotten Intervention Program (VIP) [22]. The VIP has offered free risk-factor screening and health counseling annually since 1990 to all residents in Västerbotten County within routine primary care when they turn 40, 50, and 60 years old. The VIP has previously been evaluated and described in detail [23,24].

To be eligible for participation in the VIPVIZA trial, the following inclusion criteria (defined differently according to the participant’s age groups) were used: For the 40-year group, a first-degree relative with a cardiovascular disease history before 60 years of age was considered a risk factor. For the 50-year group, those with any of the following were included: hypertension, LDL cholesterol > 4.5 mmol/L, waist circumference > 102 cm for men and >88 cm for women, smoking, diabetes, or a first-degree relative with CVD disease before 60 years of age. For the 60-year group, age was the inclusion criterion. The exclusion criterion was significant carotid stenosis, defined as >50% luminal narrowing by NASCET. In total, 3532 participants were included from April 2013 to June 2016.

The participants were asked to complete extensive questionnaires covering socioeconomic and psychosocial conditions and underwent ultrasonography examination of the carotid arteries at inclusion. Risk factor measurements, such as of the waist circumference, weight, and blood pressure, were measured at baseline for the VIP, as previously described in detail [23].

The study was approved by the Swedish Ethical Review Authority 2019-09-24 [DNR: 2011-445-31M(2019-04691)]. All participants gave their informed written consent before inclusion, and the study adhered to good clinical practice guidelines.

### 2.2. Ultrasonography/Outcomes

The two primary outcomes in this study, the presence of atherosclerotic plaques and the measurement of the carotid intima–media thickness (CIMT), were based on carotid ultrasound examinations. All carotid examinations were performed by sonographers trained in carotid ultrasound techniques and according to a standardized protocol. A portable ultrasound system with a linear 7 MHz transducer (CardioHealth Station; Panasonic Healthcare Corporation of North America, Newark, NJ, USA) was used. The presence of atherosclerotic plaques was determined according to Mannheim consensus [25], which includes assessment of the whole carotid artery (internal carotid, external carotid, bifurcation, and common carotid artery). Carotid plaques were defined as plaque absence, unilateral plaque, or bilateral plaques. The intersonographer reproducibility of carotid plaque detection has previously been evaluated in VIPVIZA, with a kappa value of 0.70 (CI 0.60–0.80) [26]. CIMT was measured in a plaque-free segment, real-time automatic cardiac peak-systole, in the distal 1 cm of the common carotid arteries bilaterally at two predefined angles at each side (120 and 150 degrees on the right side and 210 and 240 degrees on the left side). The maximum mean value of the CIMT, independent of angle and side, was used for analysis. The intersonographer variability in the CIMT has shown an intraclass correlation coefficient of 0.95 [27].

### 2.3. ABO and RhD Blood Groups

The information about ABO and RhD was retrieved by linking VIPVIZA with the Swedish–Danish database (SCANDAT-3) [28]. This was possible since all residents in Sweden are given a national registration number (NRN), allowing linkage to other registers. The SCANDAT database includes records of individuals who have received blood transfusions, donated blood, or had blood group testing done for other reasons, for example, during pregnancy. Participants registered in SCANDAT after VIPVIZA inclusion were excluded to avoid selection bias due to interference from the VIPVIZA trial on a blood typing indication. Thus, 2929 participants were included in this study, accounting for approximately 83% of all VIPVIZA participants. The proportion of eligible participants from VIPVIZA with blood group data in different age groups was 79%, 84%, and 83% in the 40-, 50- and 60-year-old groups, respectively.

### 2.4. Statistical Methods

Baseline characteristics for continuous variables are presented as the medians (25th to 75th percentiles), with percentages for categorical variables. To compare baseline characteristics in different blood groups (non-O vs. O blood groups and RhD+ vs. RhD−), we used the Mann–Whitney U-test for continuous variables and the Chi-squared test for independence for categorical variables. *p*-values of <0.05 were considered significant.

The associations between the carotid intima–media thickness (CIMT) and blood group (O vs. non-O and RhD+ vs. RhD−)were assessed by linear regression models, with the logarithmized CIMT (to base 10) as the outcome and the blood group as the independent variable, with adjustments for age and sex. Differences in the logarithmized CIMT between blood groups, with 95% confidence intervals, were retransformed to the original scale and interpreted as a ratio of geometric means.

To assess the association between carotid plaques (absent, unilateral, or bilateral plaque) and blood groups, ordinal (proportional odds) regression models were used. In multivariable analysis, adjustments were made for age and sex. The analyses for the CIMT and carotid plaques were performed and stratified by age group.

Additionally, a subgroup of participants reporting heredity for CVD was studied since blood group is a hereditary factor. Heredity was defined as having a first-degree relative with CVD before 60 years of age. In the 40-year-old group, this corresponded to all participants (inclusion criteria for VIPVIZA), while for the 50- and 60-year-old groups, we retrieved information about heredity from the VIP 10 or 20 years prior when these participants were also aged 40.

All calculations were performed with SPSS version 28 (IBM corporation, New York, NY, USA).

## 3. Results

The A, B, AB, and O distribution was 43.6%, 12.6%, 6.1%, and 37.7%, respectively. Thus, 62.3% of participants were classified as having a non-O blood group. RhD -positive participants accounted for 85.4%. When assessing the whole study population, there were no differences in baseline characteristics between O and non-O blood group individuals or RhD+ and RhD− individuals, as seen in Table 1. The baseline characteristics stratified by sex showed that smoking was more common in RhD− vs. RhD+ women; no other differences were seen (Appendix A). The presence of plaque in the carotid arteries did not differ between O and non-O or between RhD+ and RhD− participants, nor did the CIMT. The differences between excluded and included participants can be seen in a drop-out analysis in Appendix A.

The association between the blood group and CIMT can be seen in Table 2. There were no differences between non-O and O blood group individuals, neither when investigating all participants nor when stratifying by age group. For RhD, the 40-year-old RhD− individuals had increased CIMT compared to RhD+ participants. For other age groups, no significant differences were seen. When further assessing the influence of RhD on the CIMT for O and non-O blood groups separately, 40-year-old RhD− participants still showed an association with the CIMT in both groups (Appendix A).

In Table 3, the odds ratios for having carotid plaques can be seen. There was no increase in odds for non-O individuals compared to O individuals, nor was there any association observed between the RhD blood group and plaques.

Heredity for CVD (reported in a questionnaire when they were 40 years old) was associated with increased odds for carotid plaques in the 60-year-olds, with an OR of 1.33 (CI 95% 1.03; 1.71). The corresponding odds ratio for the 50-year-olds was 1.32 (CI 95% 0.89; 1.97).

The associations between blood groups and carotid plaques for individuals with heredity for CVD in the two oldest age groups are presented in Table 4. In this sub-group, non-O participants aged 60 had significantly decreased odds for carotid plaques. This was not seen in the 50-year-old group. For RhD, no association with plaques was seen, nor did we find associations between any blood group and the CIMT (Appendix A).

## 4. Discussion

In this study, we found an association between the RhD blood group and subclinical atherosclerosis in 40-year-olds, represented by RhD negative individuals having an increased CIMT compared to RhD+ individuals. In addition, we found a decreased risk of having carotid plaques in the non-O blood groups for individuals who were 60 years old and reported heredity for CVD.

To the best of our knowledge, this study is among the first to examine the influence of ABO and RhD on subclinical atherosclerosis, representing a population with low to intermediate risk for cardiovascular disease. Previous studies assessed high-risk patients with suspected or known CVD, stating that non-O individuals have an increased risk for coronary artery disease (CAD) [15,16] and more complex atherosclerotic lesions [29]. One study on low-risk individuals reported that blood group A was associated with a higher coronary artery calcium score (CACS), indicative of subclinical disease [30]. However, there have been no studies based on ultrasonography results.

For RhD status, previous studies are scarce, and the link to health outcomes is less clear. Some large-scale registry studies based on ICD codes have been conducted; however, they have not been conducted on low-risk individuals. One study reported no association with CVD [7], and another found RhD negative individuals to be associated with increased risk for iliac aneurysm but not for other cardiovascular events, despite including over 5 million individuals [5]. In general, previous studies investigating CVD risk have only considered the ABO system and not RhD.

The mechanism for the association of RhD− individuals with subclinical atherosclerosis might be related to the fact that lipoproteins have been shown to differ between RhD groups but not between ABO groups [13]. However, one study reported a better lipid profile in RhD− compared to RhD+ individuals [14], which contradicts our results. Another mechanism that may contribute to the increased risk for subclinical atherosclerosis observed in RhD− individuals is hypertension. While no statistically significant differences were seen between the groups in this study, 50.9% of RhD+ participants had hypertension, compared to 55.4% of RhD− individuals. However, this is also contradictory to the few available studies including the RhD blood group, since they found that RhD+ individuals have an increased risk for primary hypertension [5].

Our finding was only seen in 40-year-olds, who are included in the VIPVIZA study based on heredity for CVD only. However, this could not be replicated in the sub-analysis including 50- and 60-year-olds with heredity for CVD. A limitation connected to this is that the 40-year-old group has fewer participants, especially participants who have developed carotid plaques. Still, we think there is an indication that RhD has a role in the development of atherosclerosis in young individuals with a hereditary component.

For the ABO blood groups, we could not find a clear association with subclinical atherosclerosis when assessing the entire study population. Previous studies on the relationship between ABO and cardiovascular events, including several large studies and meta-analyses, have concluded that non-O individuals have an increased risk for thromboembolic events [5,6,7,8,9,10,11,31]. The literature is more ambiguous about more stable forms of atherosclerotic disease (e.g., CAD) [10,11,12]. Since some, but not all, have shown the non-O blood group to be associated only with concomitant CAD and MI and not with CAD alone [10,11]. One can therefore hypothesize that the ABO blood group system affects the later stages of atherosclerosis, i.e., plaque rupture and thrombus formation, and not the early development of atherosclerosis assessed in this study. Thus, it is not unreasonable that we found blood group O individuals aged 60 with heredity for CVD to have increased odds for carotid plaques, since they can be expected to have the highest atherosclerotic burden.

Further supporting the hypothesis that ABO affects later stages of atherosclerosis is the fact that the ABO blood group has been shown to impact the activation of platelets in the presence of existing coronary atherosclerosis [32,33]. Blood group O is also associated with lower levels of circulating von Willebrand factor (vWF) and coagulation factor VIII (FVIII) compared to other blood groups, affecting the coagulation pathway, which is also necessary for plaque rupture and concomitant thrombosis [34,35].

Although the presence of carotid plaques and an increased CIMT are associated with future CVD in large epidemiological studies [2,19], it may not consistently demonstrate the individuals overall atherosclerotic burden [36] or the plaques rupture tendency. ABO has been reported to be related to plaque characteristics, where the non-O blood groups showed plaques that were more prone to rupture. Still, the incidence of lipid plaques did not differ between ABO blood groups [37], supporting our null finding with subclinical atherosclerosis. Further investigations of plaque characteristics in MI patients showed that the non-O blood groups had a higher thrombus burden, while the O blood group still had more extensive atherosclerosis [38], which is consistent with our results in 60-year-olds with heredity that showed an association with carotid plaques.

There are some limitations to this study. One is the focal nature of atherosclerosis, and since plaque characteristics can differ between blood groups, a multi-view carotid ultrasound variable, which is more strongly associated with cardiovascular risk [39], could have been used rather than the plaque presence and single CIMT measures that were used in this study. In the current study, only the carotid artery was examined. A multi-territorial approach including aortic, iliofemoral, and coronary territories could have more accurately reflected the subclinical atherosclerotic burden [2], as well as other ultrasonography methods to assess subclinical atherosclerosis [36]. However, a strength is that we had trained sonographers and adhered to extensive protocols, including specified angles at which the CIMT is measured, to fully evaluate the degree of atherosclerotic burden and available studies on reproducibility [26,27]. Another limitation is the number of participants; consequently, there was no possibility of evaluating the ABO blood groups separately. In this study, we compared the non-O blood groups with the O-blood group, whereas other larger studies could compare the blood groups separately. This might be why these studies found an association to both atherosclerosis and myocardial infarction in blood group A compared to blood group O [5,6], while other studies comparing non-O with O mostly found an association with MI. This association was also seen in the single study assessing low-risk individuals, where a relationship between blood group A and higher CACS was found [30]. Still, after considering these limitations, we believe that our findings generate further understanding of the atherosclerotic pathophysiology. Since our findings were only seen in a small subgroup of individuals (40-year-olds and 60-year-olds with heredity), further studies including more individuals are needed. This would also make it possible to stratify results for several blood groups and not only assess non-O and O blood groups. Studies investigating the possible mechanisms, such as dyslipidemia, behind subclinical atherosclerosis in different blood groups are also scarce. However, if the findings can be replicated, we believe that blood groups could aid in risk stratification and, consequently, improve CVD prevention.

## 5. Conclusions

In conclusion, we found the hereditary RhD factor to be associated with subclinical atherosclerosis, represented by an increased CIMT in younger RhD− individuals, indicative of the RhD blood group system as a mediator in atherosclerosis. For ABO, we found no association for the whole study population. There was an increased risk for carotid plaques in 60-year-old individuals with heredity and blood group O, with the expected highest atherosclerotic burden.

## Figures and Tables

**Table 1 jcm-13-01333-t001:** Baseline characteristics stratified by blood group.

	O	Non-O	*p*-Value ^1^	RhD+	RhD−	*p*-Value ^1^
N	1102	1827		2501	428	
Male N (%)	469 (42.6)	778 (42.6)	1.00	1056 (42.2)	191 (44.6)	0.38
Age [years]	59.9 (50.0; 60.0)	59.9 (50.1; 60.0)	0.51	59.9 (50.0; 60.0)	59.9 (50.1; 60.0)	0.44
BMI [kg/m^2^]	26.9 (24.3; 30.5)	27.2 (24.3; 30.5)	0.53	27.0 (24.3; 30.5)	27.3 (24.1; 30.4)	0.79
Waist [cm]	95.0 (87.0; 105.0)	96.0 (87.0; 105.0)	0.55	96.0 (87.0; 105.0)	96.0 (88.0; 106.0)	0.55
SBP [mmHg]	128.0 (118.0; 138.0)	129.0 (118.8; 140.0)	0.22	129.0 (118.0; 139.0)	128.0 (117.3; 139.0)	0.76
Hypertension N (%)	554 (50.4)	954 (52.3)	0.34	1271 (50.9)	237 (55.5)	0.10
Diabetes N (%)	77 (7.1)	132 (7.3)	0.91	175 (7.1)	34 (8.1)	0.55
Smoking N (%)	134 (12.2)	241 (13.2)	0.43	310 (12.4)	65 (15.2)	0.13
Snuff use ^2^ N (%)	179 (16.5)	323 (18.0)	0.34	418 (17.0)	84 (20.0)	0.16
Using statin N (%)	126 (11.9)	218 (12.4)	0.73	292 (12.1)	52 (12.6)	0.86
CIMT [mm]	0.71 (0.63; 0.82)	0.70 (0.63; 0.82)	0.95	0.71 (0.63; 0.82)	0.70 (0.63; 0.81)	0.59
Plaque absence (%)	614 (55.8)	1025 (56.2)		1389 (55.6)	250 (58.4)	
Plaque unilateral (%)	279 (25.2)	430 (23.5)		615 (24.6)	94 (22.0)	
Plaque bilateral (%)	209 (19.0)	371 (20.3)	0.46	496 (19.8)	84 (19.6)	0.45

Continuous variables are presented as medians (25th to 75th percentiles), and categorical variables are presented as numbers (per cent). ^1^ Calculated with the Mann–Whitney U-test for continuous variables and Chi^2^-test for categorical variables. ^2^ Smokeless oral tobacco.

**Table 2 jcm-13-01333-t002:** Comparison of ABO blood groups (non-O vs. O) and RhD (RhD− vs. RhD+) and their association with CIMT [B (CI 95%)]. Presented in all participants and stratified by age group.

CIMT	O	Non-O	RhD+	RhD−
All age groups (N)	1102	1827	2501	428
B (CI 95%)	ref	1.00 (0.99; 1.01)	ref	0.99 (0.97; 1.01)
B (CI 95%) ^1^	ref	1.00 (0.98; 1.01)	ref	0.99 (0.97; 1.01)
40-year-old (N)	93	126	185	34
B (CI 95%)	ref	1.02 (0.98; 1.06)	ref	**1.09 (1.03; 1.14)**
B (CI 95%) ^2^	ref	1.02 (0.98; 1.06)	ref	**1.08 (1.03; 1.14)**
50-year-old (N)	302	520	707	115
B (CI 95%)	ref	0.99 (0.97; 1.02)	ref	0.98 (0.95; 1.02)
B (CI 95%) ^2^	ref	0.99 (0.97; 1.02)	ref	0.98 (0.95; 1.01)
60-year-old (N)	707	1181	1609	279
B (CI 95%)	ref	1.00 (0.98; 1.01)	ref	0.99 (0.96; 1.01)
B (CI 95%) ^2^	ref	1.00 (0.98; 1.01)	ref	0.98 (0.96; 1.01)

^1^ Adjusted for sex and age. ^2^ Adjusted for sex. B = Unstandardized B. B is interpreted as the ratio of the geometric mean of CIMT compared to the reference. Thus, B > 1 implies that the mean CIMT was higher among Non-O or RhD− compared to O or RhD+, respectively. Bold values denote statistical significance.

**Table 3 jcm-13-01333-t003:** Odds ratios [OR (CI 95%)] for carotid plaques comparing ABO blood groups (non-O vs. O) and RhD (RhD− vs. RhD+) in ordinal regression models. Presented in all participants and stratified by age group.

Carotid Plaques	O	Non-O	RhD+	RhD−
All age groups (N)	1102	1827	2501	428
Plaque presence N (%)	488 (44.3)	801 (43.8)	1111 (44.4)	178 (41.6)
Unadjusted	ref	1.01 (0.87; 1.17)	ref	0.91 (0.74; 1.12)
Adjusted ^1^	ref	0.99 (0.85; 1.15)	ref	0.88 (0.72; 1.10)
40-year-old (N)	93	126	185	34
Plaque presence N (%)	13 (14.0)	16 (12.7)	22 (11.9)	7 (20.6)
Unadjusted	ref	0.90 (0.41; 1.97)	ref	2.07 (0.82; 5.23)
Adjusted ^2^	ref	0.90 (0.41; 1.97)	ref	2.07 (0.82; 5.24)
50-year-old (N)	302	520	707	115
Plaque presence N (%)	95 (31.5)	153 (29.4)	214 (30.3)	34 (29.6)
Unadjusted	ref	0.97 (0.71; 1.31)	ref	1.03 (0.67; 1.57)
Adjusted ^2^	ref	0.96 (0.71; 1.31)	ref	1.01 (0.66; 1.55)
60-year-old (N)	707	1181	1609	279
Plaque presence N (%)	380 (53.7)	632 (53.5)	875 (54.5)	137 (49.1)
Unadjusted	ref	1.00 (0.84; 1.19)	ref	0.82 (0.65; 1.04)
Adjusted ^2^	ref	1.00 (0.84; 1.19)	ref	0.81 (0.64; 1.03)

^1^ Adjusted for sex and age. ^2^ Adjusted for sex.

**Table 4 jcm-13-01333-t004:** Odds ratios [OR (CI 95%)] comparing ABO blood groups (non-O vs. O) and RhD (RhD− vs. RhD+) in ordinal regression models. Presented in 50- and 60-year-old participants reporting heredity for CVD when aged 40 years.

Carotid Plaques	O	Non-O	RhD+	RhD−
50-year-old (N)	63	90	133	20
Plaque presence N (%)	19 (30.2)	31 (34.4)	43 (32.3)	7 (35.0)
Unadjusted	ref	1.30 (0.66; 2.58)	ref	1.13 (0.43; 2.96)
Adjusted ^1^	ref	1.36 (0.68; 2.70)	ref	1.08 (0.41; 2.86)
60-year-old (N)	89	163	220	32
Plaque presence N (%)	58 (65.2)	84 (51.5)	127 (57.7)	15 (46.9)
Unadjusted	ref	** 0.55 (0.34; 0.89) **	ref	0.72 (0.36; 1.45)
Adjusted ^1^	ref	** 0.54 (0.33; 0.88) **	ref	0.65 (0.32; 1.33)

^1^ Adjusted for sex. Bold values denote statistical significance.

## Data Availability

The data presented in this study are available on reasonable request from the corresponding author. The data are not publicly available due to the fact that the information could compromise the privacy of research participants.

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
