# Peer review of "ABO Blood Groups, RhD Factor and Their Association with Subclinical Atherosclerosis Assessed by Carotid Ultrasonography"

_jcm, 2024, doi:10.3390/jcm13051333_

Round 1

Reviewer 1 Report

Comments and Suggestions for Authors

I have received for review an original research article entitled “ABO Blood Groups, RhD-Factor and their Association with Subclinical Atherosclerosis Assessed by Carotid Ultrasonography” prepared by Malin Mickelsson et al., submitted to Journal of Clinical Medicine. Cardiovascular diseases are one of the most important public health problems in the world because, together with cancer, they cause a significant portion of deaths in many countries around the world. Although the treatment of overt cardiovascular disease has made significant progress thanks to the development of pharmacotherapy and the development of surgical and percutaneous treatment techniques, the development of knowledge about subclinical dysfunction of the cardiovascular system, methods of its assessment, and factors in which it is associated, remains a very important and current problem. Scientific research in this area may contribute to the diagnosis of cardiovascular diseases at an earlier stage. The research conducted by the Authors of the manuscript is therefore extremely important and up-to-date. I believe that the manuscript is prepared at a good level and should ultimately be considered for publication, however, some important changes are necessary, the proposals of which I would like to present below.

1)     I think the introduction is written quite well because it is informative but not too long and not overloaded with information. The Authors briefly presented why cardiovascular diseases are such an important problem, and also justified why they took up such a topic by pointing out that the topic of examining the relationship between blood type and cardiovascular disease and cardiovascular risk has already been undertaken in the past. They also pointed to the concept of subclinical atherosclerosis and mentioned carotid artery ultrasound as a method that can be used to assess a patient for subclinical atherosclerosis. I believe it is worth mentioning here that there are also other methods of assessing a patient for subclinical dysfunction of the cardiovascular system, commonly used in research, such as assessing arterial stiffness by measuring the pulse wave velocity (10.1097/HJH.0000000000002081), as well as assessing endothelial function using the flow- mediated dilation (10.3390/ijerph191811242).

2)     Although the text is written quite carefully, there are minor linguistic and editorial errors. The text should be checked again in this respect and corrected. For example, on line 90 there is "I nformed", it should be "informed". When describing the CIMT measurement methodology, it should be written what ultrasound machines were used and what transducers were used.

3)     In the table 1. there is “Snus”. It should be explained.

4)     All limitations of the study should be discussed. I think it is also a limitation that subclinical atherosclerosis was assessed only in the carotid arteries, without other vascular beds.

5)     I believe that the discussion should indicate the directions of further research (clinical and preclinical), which, in the opinion of the Authors of the paper, may further explain the importance of blood type in the pathogenesis of atherosclerosis.

Comments on the Quality of English Language

The level of English is good. The text is written understandably and generally correctly. Minor linguistic and editorial corrections are needed.

Reviewer 2 Report

Comments and Suggestions for Authors

The manuscript submitted for review is devoted to the study of such markers of subclinical atherosclerosis as blood type and Rh factor, whether the influence of these markers is associated with heredity, as well as the use of the acquired knowledge to find new preventive measures.

 The article analyzes numerous previous studies regarding predicting the risk of developing atherosclerosis, as well as significant chronic non-infectious diseases using new markers.

 The manuscript submitted for review is clear and presented in a well-structured manner. The research methods are described in detail and clearly. The data obtained as a result of the study is clearly presented in tables. The statistical analysis of the data is obvious.

 Despite some limitations noted by the authors, the results of the study are an interesting addition to the existing data on the problem under study. The conclusions are logically consistent with the presented results.

 The presented data may be of interest both to a professional audience regarding new markers of atherosclerosis risk, and to a wide range of journal readers.

 However, there are some comments that require correction.

 1. Many links are more than 5-10 years old and need to be updated.

  2. Perhaps the authors should point out the direct practical application of the acquired knowledge to medical workers and health care managers when planning activities at the individual and population levels.

Round 2

Reviewer 1 Report

Comments and Suggestions for Authors

I have received for review a revised version of the original research article entitled “ABO Blood Groups, RhD-Factor and their Association with Subclinical Atherosclerosis Assessed by Carotid Ultrasonography” prepared by Malin Mickelsson et al., submitted to Journal of Clinical Medicine. The paper has been significantly improved. I have no further critical comments. Thank you so much for the invitation to review this valuable manuscript. Congratulations to the Authors and good luck in the further scientific work.